# Effect of the Probiotic *Bacillus subtilis* DE-CA9^TM^ on Fecal Scores, Serum Oxidative Stress Markers and Fecal and Serum Metabolome in Healthy Dogs

**DOI:** 10.3390/vetsci10090566

**Published:** 2023-09-11

**Authors:** Karin Allenspach, Chi-Hsuan Sung, Jose Joaquin Ceron, Camila Peres Rubio, Agnes Bourgois-Mochel, Jan S. Suchodolski, Lingnan Yuan, Debosmita Kundu, Joan Colom Comas, Kieran Rea, Jonathan P. Mochel

**Affiliations:** 1Department of Veterinary Clinical Sciences, College of Veterinary Medicine, Iowa State University, Ames, IA 50010, USA; abmochel@iastate.edu; 2The Gastrointestinal Laboratory, Veterinary Medicine and Biomedical Sciences, Texas A&M University, College Station, TX 77843, USA; csung@cvm.tamu.edu (C.-H.S.); jsuchodolski@cvm.tamu.edu (J.S.S.); 3Department of Clinical Pathology, College of Veterinary Medicine, University of Murcia, 30100 Murcia, Spain; jjceron@um.es (J.J.C.); camila.peres@um.es (C.P.R.); lynn94@iastate.edu (L.Y.); 4Department of Biomedical Sciences, College of Veterinary Medicine, Iowa State University, Ames, IA 50010, USA; dkundu@iastate.edu (D.K.); jmochel@iastate.edu (J.P.M.); 5ADM Cork H&W Limited, Bioinnovation Unit, Food Science Building, College Road, University College Cork, T12 Y337 Cork, Ireland; joan.arias@adm.com (J.C.C.); kieran.rea@adm.com (K.R.)

**Keywords:** probiotics, *Bacillus subtilis*, microbiome, oxidative stress, metabolome

## Abstract

**Simple Summary:**

This study evaluated the daily administration of 1 × 10^9^ cfu *Bacillus subtilis* probiotics (DE-CA9^TM^) on general health, fecal scores, microbiome, as well as fecal and serum metabolome in 10 healthy adult Beagle dogs. Our data suggest that this probiotic is safe to administer to healthy Beagle dogs. Furthermore, DE-CA9^TM^ administration had a potential positive effect on several markers of oxidative stress, in addition promoting an increase in serum tryptophan levels in supplemented dogs.

**Abstract:**

Background: There is increasing interest in the use of Bacillus species as probiotics since their spore-forming ability favors their survival in the acidic gastric environment over other probiotic species. The subsequent germination of *B. subtilis* to their vegetative form allows for their growth in the small intestine and may increase their beneficial effect on the host. *B. subtilis* strains have also previously been shown to have beneficial effects in humans and production animals, however, no reports are available so far on their use in companion animals. Study design: The goal of this study was therefore to investigate the daily administration of 1 × 10^9^ cfu DE-CA9^TM^ orally per day versus placebo on health parameters, fecal scores, fecal microbiome, fecal metabolomics, as well as serum metabolomics and oxidative stress markers in ten healthy Beagle dogs in a parallel, randomized, prospective, placebo-controlled design over a period of 45 days. Results: DE-CA9^TM^ decreased the oxidative status compared to controls for advanced oxidation protein products (AOPP), thiobarbituric acid reactive substances (TBARS) and reactive oxygen metabolites (d-ROMS), suggesting an antioxidant effect of the treatment. Fecal metabolomics revealed a significant reduction in metabolites associated with tryptophan metabolism in the DE-CA9^TM^-treated group. DE-CA9^TM^ also significantly decreased phenylalanine and homocysteine and increased homoserine and threonine levels. Amino acid metabolism was also affected in the serum metabolome, with increased levels of urea and cadaverine, and reductions in *N*-acetylornithine in DE-CA9^TM^ compared to controls. Similarly, changes in essential amino acids were observed, with a significant increase in tryptophan and lysine levels and a decrease in homocysteine. An increase in serum guanine and deoxyuridine was also detected, with a decrease in beta-alanine in the animals that ingested DE-CA9^TM^. Conclusions: Data generated throughout this study suggest that the daily administration of 1 × 10^9^ cfu of DE-CA9^TM^ in healthy Beagle dogs is safe and does not affect markers of general health and fecal scores. Furthermore, DE-CA9^TM^ administration had a potential positive effect on some serum markers of oxidative stress, and protein and lipid metabolism in serum and feces.

## 1. Introduction

The use of probiotic supplementation strategies in dogs is ever-increasing, with the most common being *Lactobacillus*, *Bifidobacterium* and *Enterococcus* strains. However, there is increasing interest in the use of Bacillus species as probiotics since their spore-forming ability favors their survival in the acidic gastric environment over other probiotic species [1,2,3,4,5]. The subsequent germination of *B. subtilis* to their vegetative form allows for the growth of the probiotic bacteria in the small intestine and is believed to increase their beneficial effect on the host [2,6]. For example, some of these effects have been attributed to the secretion of enzymes, antioxidants, vitamins and anti-microbial peptides in the small intestine, which are all functions of the bacteria that are triggered by nutritional signals [4]. In addition, *B. subtilis* is known for its flexible metabolism, which can be adapted to digest carbohydrates, proteins, and lipids, depending on the availability of nutrients in the digesta [4]. This could be beneficial if there are large changes in macronutrient composition, as these bacteria may have an ameliorating effect on sudden diet changes.

Other *B. subtilis* strains have also previously been shown to have beneficial effects on digestive health and the immune system in humans and production animals [1,2,3,5,7,8,9,10,11,12,13,14,15,16,17], however, no studies have so far investigated its effects on the microbiome or metabolome in healthy dogs. This is critically needed to prove the safety of a probiotic strain in a novel species before further clinical testing in animals with gastrointestinal distress is performed. In addition, it is important to note that the analysis of the microbiome by 16srRNA sequencing methods alone may not portray the full potential benefits of probiotics since these techniques merely detect the numbers of species in the samples analyzed. These analyses therefore neglect to investigate the functional effects of probiotics in the intestine, which can be better assessed through comprehensive fecal and serum metabolomics, which represent changes in the metabolic activity of the microbes, of the gut, and of the host, respectively [18,19,20].

Furthermore, overall increased markers of oxidative stress along with diminished markers of antioxidative activity play an important role in many chronic disorders in humans and animals [21]. For example, in human patients affected with Inflammatory Bowel Disease, markers of oxidative stress have been found to be consistently increased in the serum and have therefore been identified as a hallmark of the systemic effect of this disease on the patient’s health [22]. In dogs, markers of oxidative stress have been identified to be increased in numerous disease states, such as heart disease, systemic infectious diseases (e.g., Ehrlichia canis infection and leishmaniasis) and atopic dermatitis [23,24,25]. Moreover, oxidative stress has been found to contribute to the complex pathogenesis of IBD in dogs [26] and can be reversed back to homeostatic conditions after successful therapeutic management [27].

The goal of this study was therefore to investigate the administration of 1 × 10^9^ Colony Forming Units (cfu) of the probiotic strain DE-CA9^TM^ (live *Bacillus subtilis*) orally per day versus a placebo on Complete Blood Count, biochemistry profiles, canine pancreatic lipase (cPLI), fecal scores, fecal metabolomics, as well as serum metabolomics and oxidative stress markers in ten healthy Beagle dogs over a period of 45 days.

## 2. Materials and Methods

### 2.1. Study Dogs

Ten healthy Beagles, five neutered males and five spayed females at two years of age, were kept in male–female pairs per pen at Iowa State University’s (ISU) Laboratory Animal Resources Facility in accordance with IACUC and USDA Animal Welfare guidelines.

This study was approved by and performed in accordance with the Animal Use and Care and Use Committee at ISU (IACUC protocol #20-034). Dogs were fed maintenance energy requirements of a complete commercial diet for adult dogs (Royal Canin Beagle^®^) and water was supplied ad libitum. Body condition scores and body weights were measured in all dogs on a weekly basis and were stable over the duration of the trial. The maximum amount of blood taken at any one time-point per dog was 27 mL per dog, as approved per IACUC protocol for this study.

### 2.2. Study Design

This study was a prospective placebo controlled, randomized trial with parallel design, including six healthy Beagle dogs receiving daily probiotics and four dogs receiving placebo. Test items (probiotics or placebo daily with food) were administered for a period of 45 days. Serum and fecal samples were collected on Day 0 (before treatment), Day 22, Day 35, and Day 45, and then kept at −80 °C until further analysis. Fecal scores were evaluated daily and grouped per pen according to the Purina 7-scale fecal score [28].

Serum collection: Blood was collected in sterile red top tubes and allowed to clot before centrifuging at a high speed (2000× *g* in centrifuge of 30 cm radius) for 10 min. Subsequently, the serum was removed, transferred into cryotubes, and stored at −80 degrees until further analysis. Serum was used for the analyses of chemistry profiles at the ISU Veterinary Clinical Pathology Laboratory, canine pancreatic lipase (cPLI) analysis at Idexx, and serum metabolomics (measured at UC Davis Metabolomics Center) and oxidative stress markers (measured at University of Murcia Veterinary Clinical Pathology Laboratory, Spain).

Complete Blood Count (CBC): Blood was collected into EDTA tubes and mixed immediately after collection and submitted to the ISU’s clinical pathology laboratory for analysis. EDTA blood was used for the CBC analysis only.

Probiotic and Placebo Treatment: Six of the dogs (3 × 2 pens) were administered a daily dose of 1 × 10^9^ Colony Forming Units (cfu) of DE-CA9^TM^ (live *Bacillus subtilis*) orally. Four of the dogs (2 × 2 pens) were administered a placebo control, containing medium chain triglycerides and low moisture rice dextrin in capsules looking identical to the ones containing the probiotic. All dogs were monitored daily for clinical signs of gastrointestinal discomfort, such as diarrhea, vomiting, abdominal pain and/or loss of appetite.

### 2.3. Preparation of Samples for Microbiome Analysis

The analysis of the fecal microbiome was performed at the Texas A&M University Gastrointestinal Laboratory using the dysbiosis index [29]. DNA extraction from each fecal sample (100 mg) was performed using the MoBio Power soil DNA isolation kit (MoBio Laboratories, Qiagen, Germantown, MD 2087, USA). The qPCR assays were performed according to a previously published protocol [30]. Briefly, qPCR was performed using SYBR green-based reaction mixtures. The total volume of the reaction was 10 μL. The final reaction mix consisted of 5 μL SsoFast EvaGreen^®^supermix (Bio-Rad Laboratories, Hercules, CA, USA), 0.4 μL each of a forward and reverse primer (final concentration: 400 nM), 2.6 μL of water, and 2 μL of normalized DNA (final concentration: 5 ng/μL). The conditions for the PCR were the following: first denaturation at 98 °C for 2 min, followed by 40 cycles of denaturation at 98 °C for 3 s, and annealing for 3 s. Melting curve analysis was carried out with the following temperatures: 95 °C for 1 min, 55 °C for 1 min, and increasing incremental steps of 0.5 °C for 80 cycles for 5 s each. Duplicates were run for each analysis. The data obtained from the qPCR were expressed as the log amount of DNA (fg) for one particular bacterial group/10 ng of total DNA that was isolated [30].

Serum was obtained from the dogs by peripheral venipuncture, as described above. Serum was divided in aliquots of 200 μL into cryotubes and frozen −80 °C within 30 min of the blood draw. Serum samples were used for later batch analyses of untargeted metabolomics in serum samples, as well as serum oxidant and antioxidant status measurements.

### 2.4. Untargeted Metabolomic Analysis of Fecal and Serum Samples

#### 2.4.1. Sample Preparation for Metabolome Analysis

After the samples were lyophilized and weighed, extraction was performed using a methanol/chloroform/water based method. 800 µL ice-cold methanol/chloroform (1:1, *v*:*v*) was added to each sample in a bead-based lysis tube (Bertin, Rockville, MD, USA). Samples were placed on a Precyllys 24 (Bertin) tissue homogenizer for 30 s at a speed of 6000 rpm. The supernatant was collected, and samples were homogenized again with 800 µL ice-cold methanol/chloroform. Subsequently, 600 µL ice-cold water was added to the samples, mixed, and centrifuged to separate the phases. The upper aqueous layer was then passed through a nylon filter (Merck Millipore, Burlington, MA, USA). Finally, 500 µL of the filtered aqueous phase was put through a 3 kDa cutoff column (Thermo Scientific, Waltham, MA, USA) and the extract from the column was collected for analysis.

#### 2.4.2. Metabolome Analysis by GC–MS

Metabolome analysis of the fecal and serum samples was performed at the West Coast Metabolomics Center (University of California, Davis, CA, USA) using a gas chromatography–time-of-flight mass spectrometry (GC-TOF MS) method. Samples were extracted using degassed acetonitrile. Internal standards, C08-C30 fatty acid methyl ethers (FAMEs), were added to the extracts. Samples were subsequently derivatized using methoxyamine hydrochloride in pyridine and then with *N*-methyl-*N*-trimethylsilyltrifluoroacetamide for the trimethylsilylation of acidic protons. Analytes were separated using an Agilent 6890 gas chromatograph (Santa Clara, CA, USA), and mass spectrometry was executed on a Leco Pegasus IV time-of-flight mass spectrometer (St. Joseph, MI, USA) according to a previously published protocol [31]. Unnamed peaks were excluded from statistical analysis, and peak height data were uploaded to MetaboAnalyst 4.0 (Xia Lab, McGill University, Montreal, QC, Canada). Finally, the filtered data were normalized using log transformation and Pareto scaling.

### 2.5. Analysis of Serum Antioxidant Status

All oxidant and antioxidant analyses were performed on serum samples that had been frozen at −80 °C within 30 min of blood collection.

The cupric reducing antioxidant capacity (CUPRAC) assay used for this study is based on the reduction of Cu^2+^ to Cu^1+^ by nonenzymatic antioxidants present in the sample [32]. The detection of CUPRAC was performed using protocol previously validated for use with canine serum [33]. Results were expressed in millimoles per liter of sample (mmol/L).

The ferric reducing ability of plasma (FRAP) assay used in the study is based on the reduction of ferric-tripyridyltriazine (Fe^3+^-TPTZ) to its ferrous (Fe^2+^) form. The determination of FRAP was performed according to previously published methodology [34]. Results are expressed in mmol/L.

The measurement of Trolox equivalent antioxidant capacity (TEAC) in this study is based on the assay protocol previously published by Arnao et al., which had been previously used on dog serum [35]. The principle of this assay is based on the enzymatic generation of 2,2′-azino-bis(3-ethylbenz-thiazoline-6-sulfonic acid) (ABTS) radicals. The reduction of ABTS by nonenzymatic antioxidants will be detectable in the sample. Results are expressed in mmol/L. 

The determination of total thiol for this study was based on the reaction of thiols present in the sample with 5,5′-dithiobis-(2-nitrobenzoic acid) (DTNB). This assay was performed according to previously described protocols [36]. Results are expressed in micromoles per liter (µmol/L). 

The measurement of paraoxonase type 1 (PON-1) for this study is based on the hydrolysis of phenylacetate into phenol in the sample. The phenol was measured as previously published for canine serum [24]. Results are expressed in international units per milliliter of sample (IU/mL).

Ferritin concentrations were measured using a immunoturbidimetric assay with polyclonal anti-human ferritin antibodies (Tina-quant Ferritin, Roche, Indianapolis, IN 46256, USA) and the use of the Olympus AU2700 analyzer (Olympus Diagnostica GmbH, Hamburg, Germany). Within-run and between-run coefficients of variation were below 5% and 11%, respectively. Measured levels of ferritin in serial dilutions of the serum samples resulted in linear regression equations with correlation coefficients close to 1.0. The results for ferritin were highly correlated (*r* = 0.991) with those of a species-specific canine ELISA assay kindly provided to the authors by Dr. Watanabe (Kitasato University, Japan). A Bland–Altman plot was performed to show proportional bias within both assays. 

The measurement of uric acid was performed using an automated spectrophotometric assay (Beckman Coulter, OSR6098). 

Catalase activity was determined according to a previously published protocol [37]. Catalase activity was expressed as units per liter of whole blood (U/L).

### 2.6. Measurement of Oxidant Status

The measurement of total oxidant status (TOS) for this study is based on a assay described in a previously published methodology [36]. The reaction is based on oxidants in the sample being able to oxidize Fe^2+^-o-dianisidine complex to Fe^3+^. Results are expressed in µmol/L.

The peroxide-activity (POX-Act) assay for this study is based on the measurement of total peroxides using a peroxide–peroxidase reaction with tetramethylbenzidine as the substrate for the chromogenic reaction [36]. The determination of POX-Act has also been previously used in canine serum [36]. Results are expressed in µmol/L.

The determination of reactive oxygen-derived compounds (d-ROMs) for this study is based on the sample’s ability to react with an acidic medium in the presence of *N*,*N*,-diethyl-para-phenylenediamine (DEPPD) and was determined according to a previously published protocols [36]. Results are expressed in Carratelli Units (U.CARR).

The measurement of advanced oxidation protein products (AOPP) for this study was performed using the principle that oxidized albumin and di-tyrosine contain cross-linked proteins, as has been previously published [38]. Results are expressed in µmol/L.

The measurement of thiobarbituric acid reactive substances (TBARS) for this study was based on the sample’s reaction to a stock of trichloroacetic acid, thiobarbituric acid, and *N*-hydrochloric acid (15% *w*/*v* trichloroacetic acid; 0.375% *w*/*v* thiobarbituric acid; 0.25 *N*-hydrochloric acid) and was performed under heat [39]. TBARS were determined using a previously published method for dog serum [39] with the use of a microplate reader (Powerwave XS, Biotek Instruments, Winusky, VT, USA). Results are expressed in µmol/L.

### 2.7. Statistical Analysis of Fecal and Serum Metabolites

Untargeted serum metabolites were log-transformed and Pareto-scaled. Comparisons between groups were made using two-way ANOVA, and within groups using two-way repeated measures ANOVA. Adjustment for multiple post hoc comparisons were made using Tukey’s tests. *p*-values for between group comparisons were adjusted using the Benjamini and Hochberg false discovery rate (FDR), with statistical significance set at q < 0.05 to account for the large number of analytes measured. *p*-values for within-group comparisons were adjusted using Bonferroni corrections, with significance set at *p* < 0.05.

Fecal metabolites were rank-transformed before statistical analysis since they did not meet normality assumptions. A linear mixed model was fitted, including time, group, and the interaction between time and group as fixed effects, as well as dogs as a random effect. Bonferroni correction was used for multiple pairwise post hoc comparisons with significance set at *p* < 0.05. All statistical analyses were performed using Metaboanalyst 5.0 (www.metaboanalyst.ca, accessed on 29 November 2021), GraphPad Prism 9 (GraphPad Software Inc., San Diego, CA, USA), and SPSS version 23.0.

### 2.8. Statistical Analysis of Fecal Scores

Fecal consistency was assessed and scored per pen according to the Purina scoring system by a single investigator throughput the study [28]. This scoring chart is used to evaluate the consistency of a dog’s stool from 1 to 7, with 1 being very hard and 7 being watery diarrhea. A score of 2 or 3 is considered normal, and all animals fell within this range throughout the study, with only 2 outliers above 3 observed randomly over time. As such, the average of the Purina score over 5 days was calculated per pen. The stool was assessed per pen to avoid having to isolate the dogs until they defaecated. Statistical analysis were performed using repeated measures ANOVA with the Sidak post hoc test using Graphpad Prism 9 (GraphPad Software Inc., San Diego, CA, USA).

### 2.9. Statistical Analysis of Oxidative Stress Markers and Microbiome Markers

Statistical analysis of the oxidative stress markers and microbiome markers was performed using two way ANOVA with Tukey’s post hoc test using Graphpad Prism 9 (GraphPad Software Inc., San Diego, CA, USA).

## 3. Results

### 3.1. Effect of DE-CA9^TM^ vs. Placebo on Physical Examination Findings, Complete Blood Counts (CBC), Serum Biochemistry, and Canine Pancreatic Specific Lipase (cPLI) Measurements

Daily physical examinations, CBC, serum biochemistry, and cPLI measurements were all within reference ranges at all timepoints, indicating that treatment with probiotics or the placebo were safe and well tolerated by the animals during the trial. There were no significant differences within or between groups. 

### 3.2. Effect of DE-CA9^TM^ vs. Placebo on Fecal Scoring

Daily fecal scoring was performed per pen using the Purina 7-point scoring system. The fecal score summary statistics broken into 5 day averages (means ± SEM) can be seen in Figure 1. There was an overall effect of time (*p* = 0.0006); however, there were neither treatment nor interaction effects, nor any significant differences within or between groups (Figure 1; Appendix A).

### 3.3. Effect of DE-CA9^TM^ on Gut Microbiota

Levels from the gut microbial community, including *Total Bacteria*, *Faecalibacteria*, *Turicibacter*, *Streptococcus*, *E. coli*, *Blautia, Fusobacteria*, and *Clostridium hiranonis*, were assessed using the Canine Dysbiosis Index method [29]. There was an effect of time for universal log DNA (*p* = 0.0069) and *Turicibacter* (*p* = 0.0168), and an effect of DE-CA9^TM^ supplementation for universal log DNA (*p* = 0.0069), *Faecalibacteria* (*p* = 0.04), and *Turicibacter* (*p* = 0.0216), but there were no interaction effects, and no post hoc significant differences were observed within or between groups at any timepoint (Figure 2). In addition, the Dysbiosis Index was within the normal reference range in all dogs during the study, and there was no significant effect of time or, nor an interaction effect identified (see Appendix A).

### 3.4. Effect of DE-CA9^TM^ vs. Placebo on Serum Markers of Oxidative Stress

A comprehensive panel of oxidative stress markers for anti-oxidant and oxidant status was measured in the serum of the dogs at baseline, and after treatment with probiotics or placebo (Figure 3). Peroxidase activity (Pox-Act), ferric oxidase activity (FOX), paraoxonase (PON-1), cupric ion reducing antioxidant capacity (CUPRAC), ferric reducing ability of plasma (FRAP), total equivalent antioxidant capacity (TEAC), thiols for chelating metals, uric acid, total oxidant status (TOS), and ferritin were assessed, and there was no significant effect of time or treatment, nor an interaction effect identified.

For reactive oxygen metabolites (d-ROMS), there was a significant effect of time (*p* = 0.0144), with a decrease in levels observed in the DE-CA9^TM^ group compared to their baseline at day 22 and day 45. Similarly for advanced oxidation protein products (AOPP) there was a significant effect of time (*p* = 0.0003) with a decrease in levels observed in the DE-CA9^TM^ group as compared to their baseline at day 22 and day 45. For thiobarbituric acid reactive substances (TBARS), there was an interaction effect (*p* = 0.002), and there was a significant decrease in levels at day 22 in the DE-CA9^TM^ group. However, there was also a significantly increased baseline level in the DE-CA9^TM^ group compared to the controls (Figure 3, Appendix A).

### 3.5. DE-CA9^TM^ Has an Overall Effect in the Fecal Metabolites Associated with Tryptophan, Lipid, and Fatty Acid Metabolism, as Well as Amino Acids

Fecal metabolomics revealed 257 named metabolites. There was a reduction in metabolites associated with tryptophan metabolism, specifically indole-3-acetate, indoxyl sulphate, kynurenic acid, kynerinine, indole-3-lactate, and indole-3-propionic acid, and in metabolites associated with lipid and fatty acid metabolism, including 2/3 hydroxibutyric acid, isomyristic acid, ethanolamine, sulphurol, glycerol, linoleate, linolenate, palmitoleic acid, oleic acid, phenylpropanoic acid, and cis-gonodic acid.

Of the amino acids, DE-CA9^TM^ significantly decreased phenylalanine and homocysteine and increased homoserine and threonine levels. There was only a moderate effect of DE-CA9^TM^ on any other metabolite groups associated with amino acid metabolism, carbohydrate metabolism, and other microbial metabolites, including aconitic acid, citric acid, hydroquinone, and isocitrate (Figure 4, Appendix A).

### 3.6. DE-CA9^TM^ Has an Overall Effect on Amino Acid, Fatty Acid, Vitamin, and Microbial Metabolism in Serum

Amino acid metabolism was also affected in the serum metabolome, with increased levels of urea and cadaverine, and reductions in *N*-acetylornithine in DE-CA9^TM^ compared to controls. Similarly, changes in essential amino acids were observed, with a significant increase in tryptophan and lysine levels and a decrease in homocystine. Indeed, tryptophan metabolites were also impacted with decreased levels of indoxyl sulphate; 2,8-dihydroxyquinoline; and kynurenic acid. DE-CA9^TM^ increased the levels of octadecylglycerol and 1,5-anhydroglucitol, with decreased glycerol, glycerate, sucrose, and cis-10-heptadecanoic acid levels as molecules connected to fatty acid and carbohydrate metabolism (Figure 5). Finally, an increase in guanine and deoxyuridine was detected, with a decrease in beta-alanine in the animals that ingested DE-CA9^TM^.

## 4. Discussion

This study investigated the daily administration of 1 × 10^9^ cfu DE-CA9^TM^ orally per day versus a placebo on health parameters, fecal scores, fecal microbiome, fecal metabolomics, as well as serum metabolomics and oxidative stress markers in ten healthy dogs in a parallel, randomized, prospective, placebo-controlled design. The limitations of this proof-of-concept study include the restricted size of our cohort, which, overall, hampers statistical power to detect significant differences between treatment groups. In addition, the trial was not designed as a cross-over study, which could have minimized inter-dog variability in outcome measures such as the fecal and serum metabolomics [40,41]. Lastly, looking in a population that had an increased oxidative stress load due to ageing, the administration of different dietary types, or metabolic syndromes may have resulted in changes in the baseline levels that could then be attenuated with the treatment strategy [42,43,44,45,46]. 

Despite the limitations mentioned above, significant differences were observed for oxidative stress markers and in both fecal and serum metabolomic data, showing an overall positive effect of the probiotic treatment over the 45 days of the study.

For serum markers of oxidative stress, we assessed levels of total oxidant status (TOS), peroxide-activity (POX-Act), reactive oxygen-derived compounds (d-ROMs), advanced oxidation protein products (AOPP), and thiobarbituric acid reactive substances (TBARS). An increase in these markers would be indicative of a pro-oxidant status. DE-CA9^TM^ decreased the oxidative status compared to the controls for AOPP, TBARS, and d-ROMS, suggesting an antioxidant effect of the treatment. For antioxidant status, there was no insult to the animals as they were all healthy dogs. As such, we did not detect any changes in antioxidant markers, including cupric reducing antioxidant capacity (CUPRAC), ferric reducing ability of plasma (FRAP), Trolox equivalent antioxidant capacity (TEAC), thiol, and paraoxonase type 1 (PON-1).

The main changes observed in the fecal metabolomic profile were in the tryptophan, amino acid, and fatty acid metabolism. The presence of DE-CA9^TM^ caused a general reduction in the tryptophan metabolites in the gut. Tryptophan is oxidized in the gut following the kynurenine degradation pathway, leading to the formation of kynurenine, kynurenic acid, and anthranilic acid, or by bacteria via the indole pathway, forming indole, indole-3-acetate, and tryptamine metabolites [46,47]. During exposure to DE-CA9^TM^, metabolites found in both pathways, namely kynurenine, kynurenic acid, anthranilate, and indole-3-acetate were significantly reduced (Figure 4). This suggests a reduction in the degradation of tryptophan in the gut, potentially facilitating its absorption or driving tryptophan use through the serotonin pathway [48,49]. Furthermore, DE-CA9^TM^ treatment changed the concentration of several amino acids in the gut. For example, phenylalanine concentrations were reduced in the gut. High concentrations of this amino acid in the gut have been correlated to increased colonic inflammation and bacterial translocation through the intestinal barrier [50,51]. Homoserine and threonine, also increased by DE-CA9^TM^, function as intermediates in the synthesis of other amino acids and play a role in glycolysis, the Kreb’s cycle, and the formation of aspartate. Homocysteine was also reduced in the intestinal lumen, possibly being converted to cystine through the trans-sulfation pathway using vitamin B12 and folate as cofactors [52]. Low levels of homocysteine have been associated with a reduced risk of atherosclerotic disorders, immune homoeostasis, and higher concentrations of cobalamin [53,54]. Another major effect of DE-CA9^TM^ intake was observed in gut lipid metabolism. Levels of ethanolamine decreased with the presence of the probiotic. An increase in metabolism of this fatty acid has been associated with a higher utilization of fatty acids and lipids, the proliferation of intestinal cells, and the regulation of inflammation [55,56,57,58]. Indeed, concentrations of dietary fats glycerol, oleic acid, palmitoleic acid, phenylpropanoic acid, isomyrisitic acid, linoleate, and linolenate, were reduced in these animals, suggesting a connection between ethanolamine consumption and increased fatty acid metabolism [59].

Some of the metabolic changes observed in the feces were also reflected in the serum metabolites. In serum, tryptophan levels were significantly increased and positively correlated with the reduction in tryptophan metabolites observed in the gut and serum (Figure 5). Tryptophan is an essential dietary amino acid that undergoes metabolism via the tryptophan–kynurenine pathway. The metabolites of tryptophan play a key role in many physiological processes, including cell growth and maintenance, immunity, disease states, and responses to environmental and dietary changes, as well in the gut–brain axis [46,60,61,62,63]. One of the benefits seen with the supplementation of this essential amino acid has been associated with an increase in serum serotonin, niacin, and protein biosynthesis [63,64], which has been implicated in the host response to various stressors.

Several other essential amino acids derived from diet were upregulated by DE-CA9^TM^ but failed to reach statistical significance. In general, protein and amino acid metabolism was increased in the dogs ingesting DE-CA9^TM^. Furthermore, the decreased lipid metabolism detected in the gut correlated with serum findings, where the saturated fat heptadecanoic acid was reduced as well as the levels of D-glycerate [59].

## 5. Conclusions

Data generated throughout this study support the notion that the daily administration of 1 × 10^9^ cfu of DE-CA9^TM^ in healthy Beagle dogs is safe and does not affect markers of general health and fecal scores. Furthermore, DE-CA9^TM^ administration had a positive effect on some serum markers of oxidative stress (i.e., AOPP, d-ROMS, and TBARS), which have previously been found to be abnormal in dogs with acute and chronic gastrointestinal disease [24,65,66]. Such possible antioxidative effects of DE-CA9^TM^ may also play a role in supporting healthy ageing [67]. Finally, the metabolic readouts suggest a role for this probiotic supplement in protein and lipid metabolism. An interesting observation was made with regards to the increased serum tryptophan in dogs supplemented with DE-CA9^TM^, which may warrant further research in possibly assessing its role as a dietary supplement to ameliorate the response to stressors (acute and chronic) or its role in healthy ageing (cognitive assessments).

## Figures and Tables

**Figure 1 vetsci-10-00566-f001:**
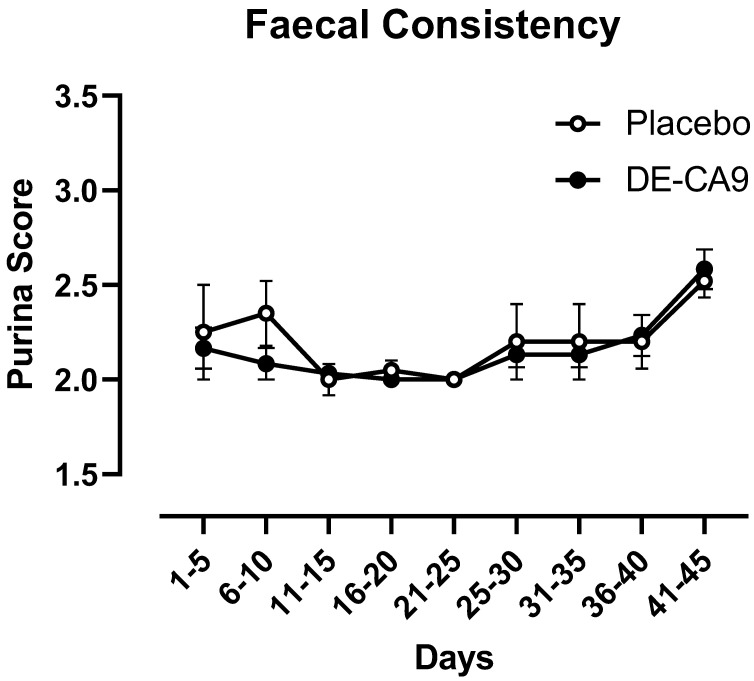
Fecal scores (scored using the Purina 7–point scoring scheme) over the treatment period (day 0–45) for placebo and DE-CA9^TM^ Treatment. All fecal scores were observed to be at scores 2 and 3, with only 2 outliers above 3 observed randomly over time. While there was an effect of time (F(8, 72) = 4.806, *p* < 0.0001), there was no effect of treatment (F(1, 72) = 0.7750, *p* = 0.3816) nor an interaction effect of time X treatment (F(8, 72) = 0.4132, *p* = 0.9094).

**Figure 2 vetsci-10-00566-f002:**
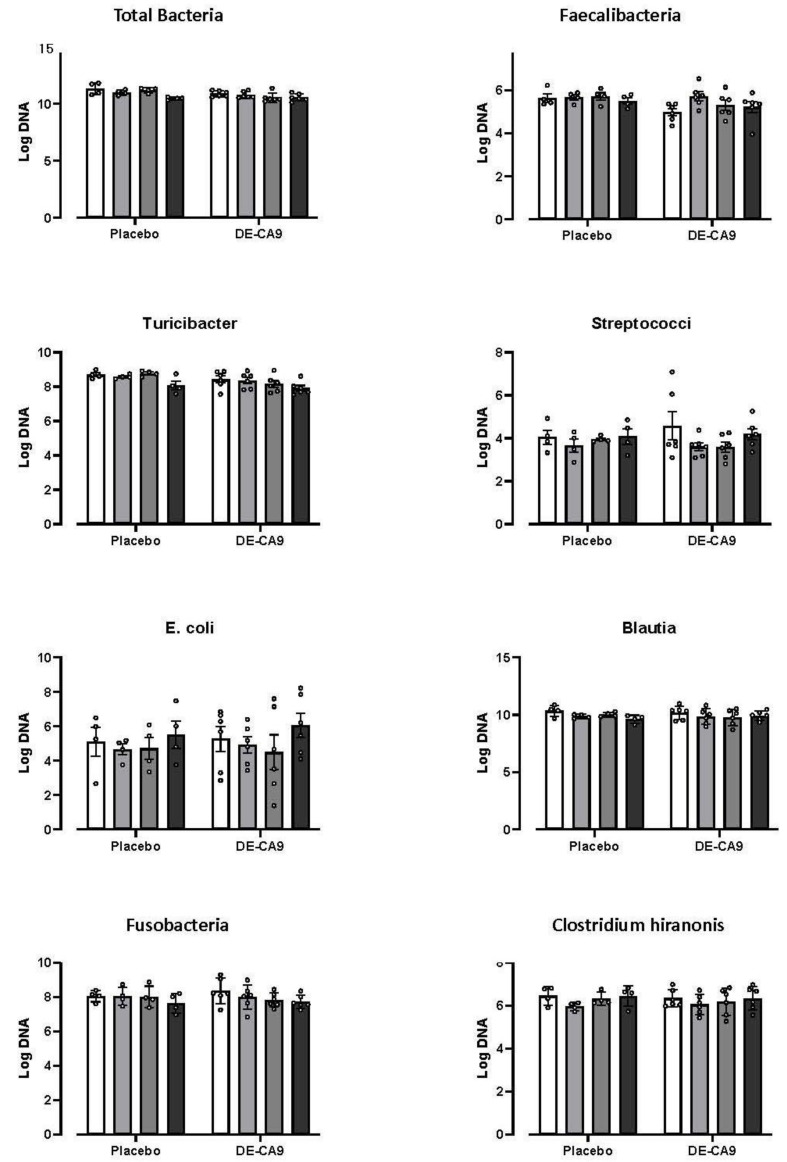
*Total Bacteria*, *Turicibacter*, *Streptococcus*, *E. coli*, *Blautia*, *Fusobacteria*, and *Clostridium hiranonis* were assessed. There was an effect of time for *Total Bacteria* (*p* = 0.0069) and *Turicibacter* (*p* = 0.0168), and an effect of DE-CA9^TM^ supplementation for universal log DNA (*p* = 0.0069), *Faecalibacteria* (*p* = 0.04), and *Turicibacter* (*p* = 0.0216), but there were no interaction effects, and no post hoc significant differences were observed within or between groups at any timepoint. Columns represent individual pens of dogs (2 dogs per pen for combined fecal samples).

**Figure 3 vetsci-10-00566-f003:**
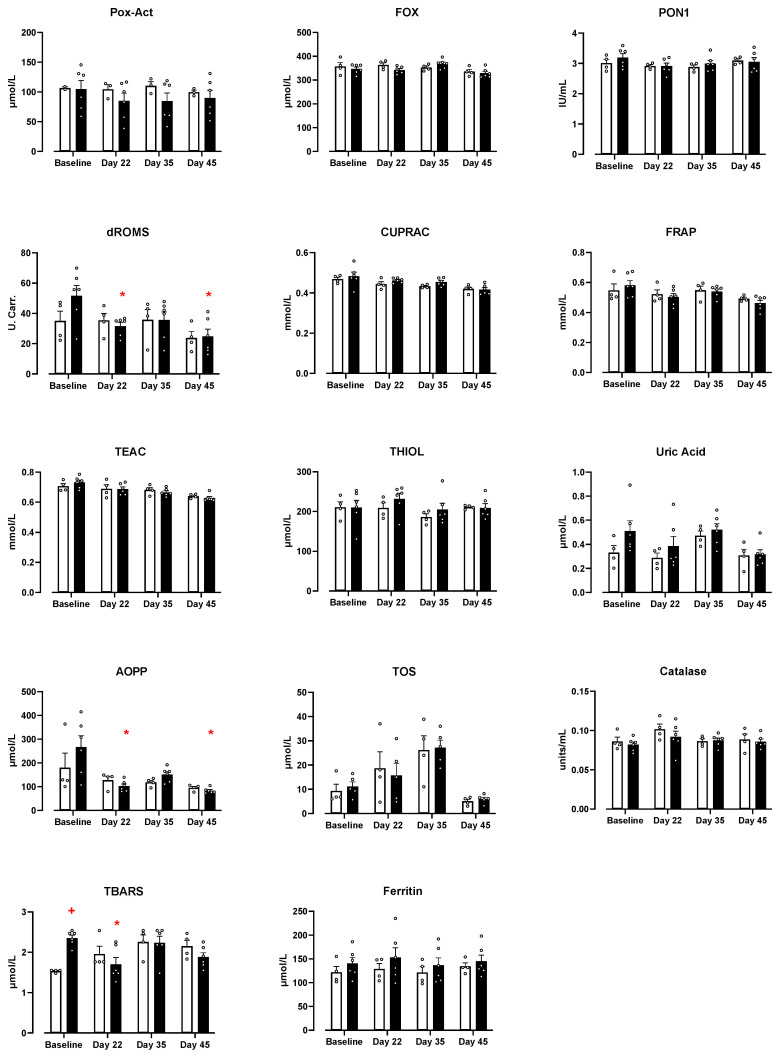
DE-CA9^TM^ significantly decreased levels of d-ROMS, TBARS, and AOPP in the serum compared to baseline, whereas no such decrease was seen in the placebo group. There was a significant effect of time for FOX (*p* = 0.0262), dROMS (*p* = 0.0144), CUPRAC (*p* = 0.0004), FRAP (*p* = 0.0125), TEAC (*p* < 0.0001), Uric acid (*p* = 0.0294), AOPP (*p* = 0.0003), and TOS (*p* < 0.0001). There was no effect of treatment per se for any group, and an interaction effect was observed for TBARS (*p* = 0.002). Black bars represent the DE-CA9^TM^ group. + represents *p* < 0.05 compared to the placebo baseline; * represents *p* < 0.05 compared to the DE-CA9^TM^ baseline.

**Figure 4 vetsci-10-00566-f004:**
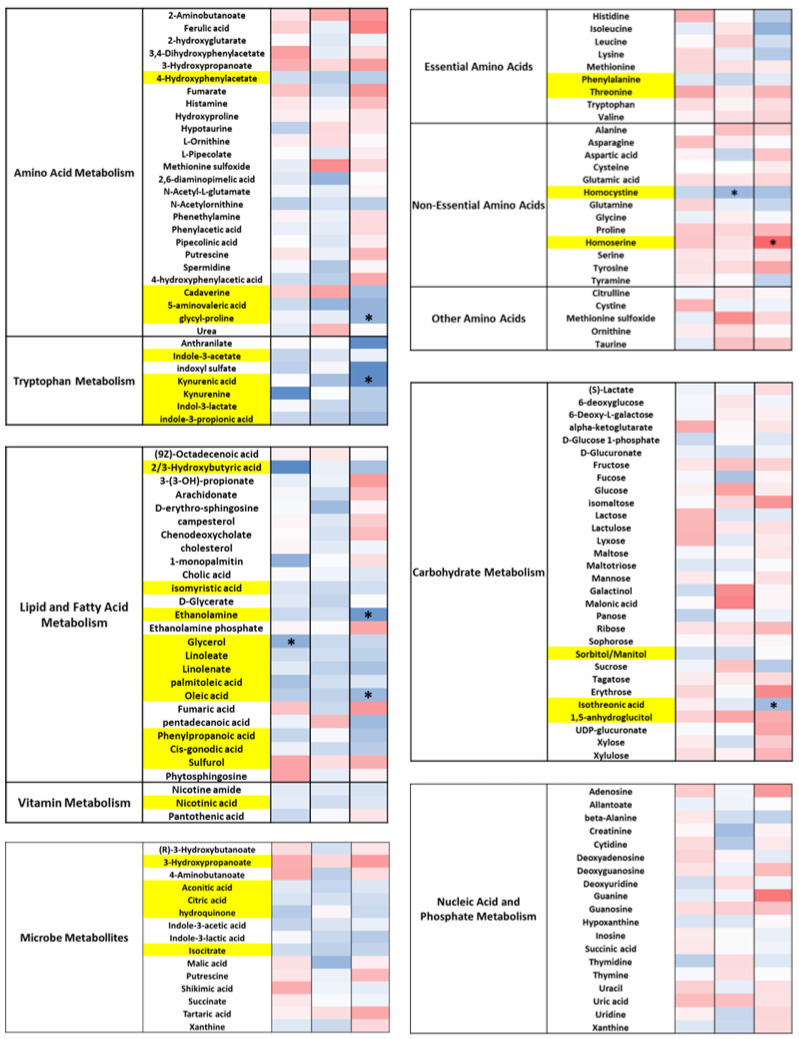
DE-CA9^TM^ decreased several metabolites associated with amino acid metabolism, lipid and fatty acid metabolism and tryptophan metabolism in feces from the canines. Yellow shading represents an overall significant effect of treatment between DE-CA9^TM^ and the placebo. * represents *p* < 0.05 between DE-CA9^TM^ and placebo at this respective timepoint.

**Figure 5 vetsci-10-00566-f005:**
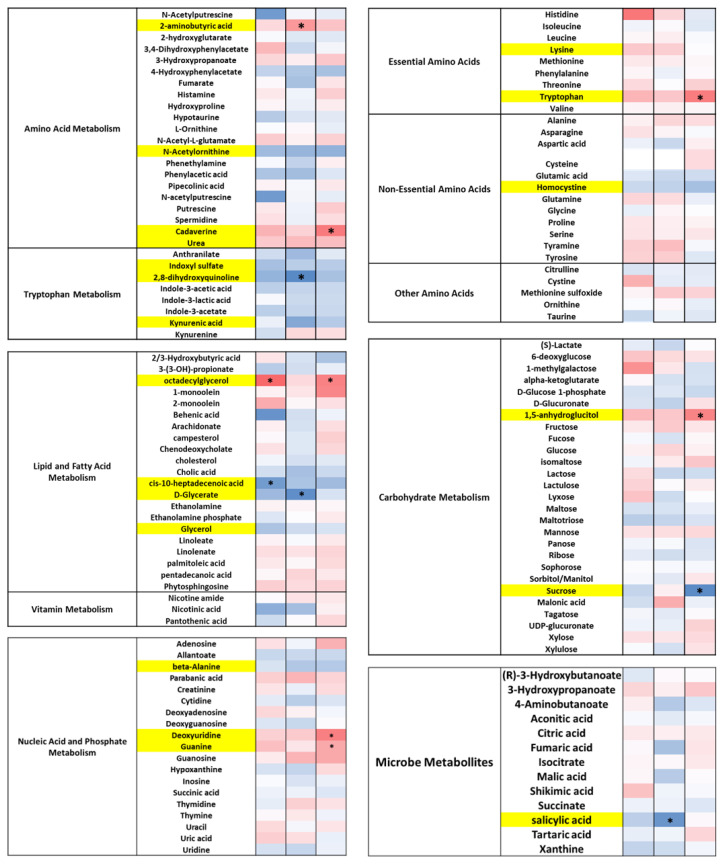
DE-CA9^TM^ decreased several metabolites associated with amino acid metabolism, lipid and fatty acid metabolism, and tryptophan metabolism in serum from the canines. Levels of 2-aminobutyric acid, cadaverine, urea, lysine, and tryptophan were significantly increased in the DE-CA9^TM^ group compared to the controls. Yellow shading represents an overall significant effect of treatment between DE-CA9^TM^ and the placebo. * represents *p* < 0.05 between DE-CA9^TM^ and placebo at this respective timepoint.

## Data Availability

The original contributions presented in the study are included in the article/Appendix A, and further inquiries can be directed to the corresponding author.

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
