# Peer review of "Effect of the Probiotic Bacillus subtilis DE-CA9TM on Fecal Scores, Serum Oxidative Stress Markers and Fecal and Serum Metabolome in Healthy Dogs"

_vetsci, 2023, doi:10.3390/vetsci10090566_

Round 1
Reviewer 1 Report
Dear Authors,
Congratulations on your research. Please find enclosed comments on the submitted manuscript:
Line 52,53,54 – the sentence is not clear so please rephrase.
Line - 96 - do you mean maintenance energy requirements (not resting)? Information about BCS is lacking. Did you follow up on the weight of the dogs?
Line 147,148,149 - not clear for what analysis you collected the serum?
Line 163 - Analysis was done by GC as you explained not HPLC
Line 241 - Serum GPox - add the full name. Not included in the results afterward? Any reason for that?
Line 303 - Dysbiosis index is not shown as an index. Could you include the results of the mixed itself?
Results - no need to write in the text p-value and f-value. If relevant you can include the absolute concentration of analysis, but that is often shown on the graph.
Figure 3. When marking the significant difference between the group please use the usual approach - marking with the * and connecting with the line the bars that are significantly different, and increase the visibility.
Line 394-396 The sentence does not make sense - as the dogs are the same age, young and healthy - that statement is not applicable. Individual differences could have an effect, which is addressed in the part when you explain the size of the cohort and the design of the study.
407 "The insult" - rephrase
Regards,
English Language should be revised.
Author Response
Rebuttal letter to reviewer 1
Dear Authors,
Congratulations on your research. Please find enclosed comments on the submitted manuscript:
Line 52,53,54 – the sentence is not clear so please rephrase.
Many thanks for this comment. We have re-phrased this paragraph to the following:
In addition, B. subtilis is known for its flexible metabolism which can be adapted to digest carbohydrates, proteins and lipids, depending on the availability of nutrients in the digesta [6]. This could be beneficial if there are large changes in macronutrient composition, as these bacteria may have an ameliorating effect on sudden diet changes.
Line - 96 - do you mean maintenance energy requirements (not resting)? Information about BCS is lacking. Did you follow up on the weight of the dogs?
Many thanks for this comment. Correct- this should mean maintenance energy requirements and this was corrected in this sentence.We did follow up on both BCS and body weight and they were all stable over the course of the trial.
This paragraphs now reads:
Dogs were fed maintenance energy requirements of a complete commercial diet for adult dogs (Royal Canin BeagleÒ) and water was supplied ad libitum. Body condition Scores and body weights were measured in all dogs on a weekly basis and were stable over the duration of the trial. The maximum amount of blood taken at any one time-point per dog was 27ml per dog, as approved per IACUC protocol for this study.
Line 147,148,149 - not clear for what analysis you collected the serum?
Many thanks for this comment.
This paragraph has now been corrected:
Serum was obtained from the dogs as described above. Serum was placed in aliquots of 200ul into cryotubes to be frozen at -80°C within 30 min of the blood draw. Serum samples were used for later batch analysis of untargeted metabolomics in serum samples, as well as serum oxidant and antioxidant status.
Line 163 - Analysis was done by GC as you explained not HPLC
Thank you very much for this comment. This has been corrected:
3.4.2 Metabolome analysis by GC–MS
Line 241 - Serum GPox - add the full name. Not included in the results afterward? Any reason for that?
Thank you very much for this comment- yes this was not measured in the trial and so this sentence was removed from the manuscript.
Line 303 - Dysbiosis index is not shown as an index. Could you include the results of the mixed itself?
Thank you very much for this comment. We have included the raw data for the dysbiosis index as well as the individaul qPCR data for each of the bacterial strains in teh supplementary data.
Results - no need to write in the text p-value and f-value. If relevant you can include the absolute concentration of analysis, but that is often shown on the graph.
Thank you for this comment. We have elected to leave the p-values in the text and figures, but have deleted the F values.
Figure 3. When marking the significant difference between the group please use the usual approach - marking with the * and connecting with the line the bars that are significantly different, and increase the visibility.
Thank you very much for this comment. We have added bars to the significant datapoints in figure 3 and increased the resolution of the graphic.
Line 394-396 The sentence does not make sense - as the dogs are the same age, young and healthy - that statement is not applicable. Individual differences could have an effect, which is addressed in the part when you explain the size of the cohort and the design of the study.
Thank you for this comment.We have re-phrased the paragraph to the following:
Data generated throughout this study support the notion that daily administration of 1x109 cfu of DE-CA9TM in healthy Beagle dogs is safe and does not affect markers of general health and fecal scores. Furthermore, DE-CA9TMadministration had a significant effect on some serum markers of oxidative stress (i.e., AOPP, d-ROMS and TBARS), which have previously been found to be abnormal in dogs with chronic gastrointestinal disease [53].
407 "The insult" – rephrase
Thank you very much for this comment.
This sentence has been re-phrased:
there was no compromise to the animals as they were all healthy dogs.
Reviewer 2 Report
I would like to congratulate an excellent work and thank you for a fascinating reading. In my opinion Your text can be considered a kind of template for further Authors - on how to master scientific reports. Best wishes!
Author Response
Thank you so much for taking the time to review our manuscript and for your kind words.